# Intermittent Theta Burst Stimulation Combined with Cognitive Training to Improve Negative Symptoms and Cognitive Impairment in Schizophrenia: A Pilot Study

**DOI:** 10.3390/brainsci14070683

**Published:** 2024-07-08

**Authors:** Alessandra Vergallito, Camilla Gesi, Sara Torriero

**Affiliations:** 1Department of Psychology & Neuromi, University of Milano-Bicocca, 20126 Milan, Italy; 2Department of Mental Health and Addictions, ASST Fatebenefratelli-Sacco, 20157 Milan, Italysara.torriero@asst-fbf-sacco.it (S.T.)

**Keywords:** schizophrenia, neurostimulation, negative symptoms, cognitive impairment, cognitive remediation, randomized controlled trial, theta burst stimulation, combined approach

## Abstract

Schizophrenia is a chronic psychiatric disorder severely affecting patients’ functioning and quality of life. Unlike positive symptoms, cognitive impairment and negative symptoms cannot be treated pharmacologically and represent consistent predictors of the illness’s prognosis. Cognitive remediation (CR) interventions have been applied to target these symptoms. Brain stimulation also provides promising yet preliminary results in reducing negative symptoms, whereas its effect on cognitive impairment remains heterogeneous. Here, we combined intermittent theta burst stimulation (iTBS) with CR to improve negative symptoms and cognitive impairment in schizophrenia spectrum patients. One hundred eligible patients were invited, and twenty-one participated. We randomized them into four groups, manipulating the stimulation condition (real vs. sham) and CR (no training vs. training). We delivered fifteen iTBS sessions over the left dorsolateral prefrontal cortex for three weeks, followed (or not) by 50 min of training. Consensus-based clinical and cognitive assessment was administered at baseline and after the treatment, plus at three follow-ups occurring one, three, and six months after the intervention. Mixed-model analyses were run on cognitive and negative symptom scores. The preliminary findings highlighted a marginal modulation of iTBS on negative symptoms, whereas CR improved isolated cognitive functions. We herein discuss the limitations and strengths of the methodological approach.

## 1. Introduction

Schizophrenia is a chronic mental disorder that significantly impairs patients’ and caregivers’ lives, resulting in substantial social and economic burdens [1,2]. The clinical symptoms of schizophrenia have been classified into three main categories: positive symptoms, negative symptoms, and cognitive impairment [3,4]. According to the Diagnostic and Statistical Manual of Mental Disorders fifth edition (DSM-5) criteria [5], positive symptoms refer to the excess or distortion of ordinary functions, such as hallucinations, delusions, and disorganized behavior. Negative symptoms, instead, concern significant reductions in or the absence of behaviors and functions. Specifically, the DSM-5 suggests that diminished emotional expression (through facial and bodily indexes, but also prosodic ones) and avolition (the inability to initiate and maintain goal-directed activities) are prominent in the disorder. Other negative symptoms include the apparent lack of interest in social interaction (asociality), diminished speech output (alogia), and the inability to experience or recall pleasure from activities or relationships [5,6,7,8]. Lastly, cognitive deficits in schizophrenia are characterized by a global impairment in social and nonsocial cognition, involving a broad range of functions that include processing speed, visuospatial and verbal episodic memory, working memory, attention, executive functions, reasoning, and decision-making [9,10,11,12,13,14,15,16], with large interindividual differences considering the severity of impairment [17]. Cognitive dysfunctions typically precede psychosis onset and persist in non-acute phases even when the other symptoms improve [11,18,19,20]. Moreover, cognitive deficits have been found in patients’ first-degree relatives, thus highlighting that cognitive deficits may represent a key trait of schizophrenia [21].

A high percentage of patients experience negative symptoms and cognitive impairment (about 40% and 80%, respectively) [22]. The two symptoms often co-occur [23] and influence patients’ prognosis, affecting treatment compliance, symptom recurrence, and functional disability in daily life [24,25].

Although antipsychotic medications represent the standard treatment for schizophrenia, they mainly act on positive symptoms, failing to modulate negative symptoms and cognitive deficits [26,27]. Therefore, alternative approaches should be considered to improve these illness features. Behavioral interventions such as cognitive remediation (CR) can be beneficial to improve or restore cognitive functioning. CR refers to evidence-based training targeting impaired cognitive functions to enhance patients’ abilities and generalize such changes to durable improvements in their everyday lives [28,29,30]. For example, enhancing sustained attention through specific exercises may be generalized to improvements in attention at school or during work-related activities. CR strategies can be distinguished into two primary approaches: restorative and compensatory [31]. Restorative strategies aim to repair impaired cognitive skills. They are based on the neuroscientific evidence of neuronal plasticity, namely the ability of the brain to change and re-organize functionally and structurally throughout the lifespan in response to experience and injury [32,33]. Within this approach, it is possible to distinguish between bottom–up and top–down interventions. Bottom-up interventions focus on restoring basic cognitive functions, such as attention, and then proceed to more complex ones, such as problem-solving. Top-down interventions, instead, aim to develop more complex abilities, such as problem-solving and working memory, under the assumption that basic components like attention and processing speed are engaged and trained simultaneously [31]. Restoring techniques may include drill and practice exercises to restore a specific cognitive function or execute different tasks based on the same function, thus promoting the generalization of new acquisitions to different contexts (see [28] for further discussion considering restoring techniques). In contrast, compensatory strategies do not try to restore impaired skills. Instead, they focus on compensating for or bypassing deficits by using the individual’s remaining cognitive abilities and/or acting on the individual’s environment. For example, external or environmental strategies may include using diaries or checklists to support memory and daily organization (see [34] for a recent meta-analysis on compensatory strategies in psychosis). Meta-analyses typically report small to moderate effect sizes of CR in reducing cognitive impairment [35] and negative symptoms [36] in schizophrenia, with large interindividual variability in improvements and generalizability to daily functioning [37]. It should be highlighted, however, that negative symptoms are more typically targeted through social skills training, cognitive behavioral therapy, and family interventions, but no robust indications in favor of a specific intervention have been reported [6,38,39]. 

At a neurophysiological level, converging evidence has highlighted widespread abnormalities at the structural and functional level in schizophrenic patients compared to healthy controls, including a reduction in gray matter volume in cortical regions, especially frontal and temporal areas, and subcortical structures, including the amygdala and hippocampus [40,41,42], and hypoconnectivity between the frontoparietal, salience, and default mode networks [43,44]. In particular, the dorsolateral prefrontal cortex (DLPFC) has been suggested as a crucial hub in the neural underpinnings of negative and cognitive symptoms [40,45]. For instance, in vivo neuroimaging studies highlighted DLPFC hypoactivity during several tasks known to recruit this region, such as working memory or cognitive control exercises (for a review, see [45]). Interestingly, similar neuroanatomical features have been reported in people at risk of psychosis, although to a lesser extent, suggesting that previous findings are not the mere consequences of antipsychotic drug treatment but may arise from genetic factors (see for a review [46]). 

In line with these findings, brain stimulation techniques, such as transcranial magnetic stimulation (TMS), have received considerable attention for improving abnormal excitability and brain connectivity [47,48]. TMS delivers a strong, short magnetic pulse to the patient’s head, which can generate neuronal firing by inducing suprathreshold neuronal membrane depolarization [49]. When applied to induce long-term effects on the targeted network, TMS pulses can be repeatedly delivered (repetitive TMS, rTMS) following specific pulse frequencies or patterns [50]. Several meta-analyses highlighted the positive effect of rTMS in reducing negative symptoms [51,52,53]. Conversely, heterogeneous results emerged from studies analyzing the rTMS effect on cognitive impairment, sometimes suggesting improvements in specific functions, such as working memory [54], sometimes reporting non-significant effects [55,56,57]. Interestingly, some studies did not find improvements immediately after the treatment but later in follow-ups [58,59,60].

It is worth noting that most of the cited studies applied stimulation or CR protocols as stand-alone treatments or as an add-on to pharmacotherapy, without combining the two. Although the use of combined treatments in psychiatry is still in its infancy, previous findings suggest the convenience of this multimodal approach, delivering stimulation before (priming effect), during (synergistic effect), or after (consolidation effect) the cognitive or behavioral intervention [61,62,63,64]. Research from experimental neuroscience points in the same direction, highlighting that the effects of non-invasive brain stimulation are state-dependent, meaning that the state of target regions plays a crucial role in modulating the effect of non-invasive brain stimulation on cortical excitability and behavioral outcomes [65,66,67,68,69]. Since previous evidence indicates that both rTMS and cognitive training modulate cortical connectivity and neuroplasticity [70], time-locking them may maximize their effectiveness [71,72,73].

### Study’s Objectives and Expected Results

The current study aims to investigate whether the application of a multimodal intervention combining iTBS with CR could improve cognitive abilities and negative symptoms in patients with schizophrenia spectrum disorders.

Based on the previously discussed evidence, we expected that CR as a stand-alone treatment would improve cognitive impairment and negative symptoms. Similarly, we hypothesized that iTBS might enhance cognitive performance and reduce negative symptoms. Crucially, we wanted to disentangle whether a multimodal approach could boost the two interventions at the treatment end (primary endpoint) and at longer time points (secondary endpoint). 

With these aims in mind, we designed an experiment in which participants underwent a three-week treatment, including fifteen sessions (once a day for 5 weekdays) in which real or sham iTBS was followed (or not) by a cognitive intervention that trained nonsocial and social cognitive functions (in the present study, we focus on the cognitive and negative symptom outcomes, whereas changes in social cognition abilities have been analyzed in another study [74]). Cognitive performance and negative symptoms were evaluated at baseline and immediately after the end of treatment, plus in three follow-ups occurring one month, three, and six months after the intervention. 

## 2. Materials and Methods

### 2.1. Participants

We conducted the a priori power analysis with G-Power software 3.1 [75] to establish the sample size required to detect an effect size of 0.36, which is the effect size reported by Aleman and colleagues’ meta-analysis [51] on non-invasive brain stimulation modulation of negative symptoms when including only studies targeting the left prefrontal regions. We imposed a power index of 0.90 and an alpha of 0.05 for 4 groups (iTBS, sham iTBS, iTBS + cognitive training, sham iTBS + cognitive training) and 5 measurements (pre- and post-treatment, plus three follow-ups). Since a repeated-measures ANOVA typically violates the sphericity assumption, we applied the most conservative sphericity correction using the formula 1/1−m, where m represents the number of measurements (see [76]—Appendix A). The study took place between July 2020 and July 2023. During the three years, a total number of 100 patients were invited to participate in the study. They were recruited through the ASST Fatebenefratelli-Sacco and Fondazione IRCCS San Gerardo dei Tintori (Italy). Patients were eligible if they met the inclusion criteria (described later) and their referring psychiatrists considered the treatment feasible for their actual clinical conditions. The study was first presented to eligible patients by their psychiatrist, and then, those who expressed interest in participating were contacted by the study’s Principal Investigator (S.T.) for an informational meeting and to finally decide whether to participate. Following the power analysis, our original plan was to include 40 patients (10 per group). However, we could not collect the foreseen sample size due to the COVID-19 pandemic and the low rate of patients agreeing to participate in the study (see Figure 1). Eventually, twenty-one participants (5 females, mean age = 35.2 ± 10.8, mean years of illness duration: 10 ± 8.8) participated in the study. 

Inclusion criteria comprised an age between 18 and 60 years, a diagnosis falling within the schizophrenia spectrum according to the criteria of the DSM-5 [5], and medication being stable for at least 3 months before study participation. Exclusion criteria comprised any contraindications to TMS procedures, such as neurological disorders or pregnancy, substance dependency six months before inclusion in the study, and the incapacity to provide informed consent for study participation. Considering that schizophrenic patients may present several psychiatric comorbidities, such as major depression, anxiety, obsessive-compulsive, and trauma-related disorders (e.g., [77]), comorbid psychiatric conditions were not considered among the exclusion criteria nor were they analyzed in more depth since we expected them to be randomly distributed across the different experimental conditions.

The local ethics committee approved the study (2018/ST/081), and participants were treated in accordance with the Declaration of Helsinki. Written informed consent was given from all participants before the study procedures began. 

Table 1 summarizes participants’ demographic and baseline clinical features. The four groups did not differ in such measures.

**Figure 1 brainsci-14-00683-f001:**
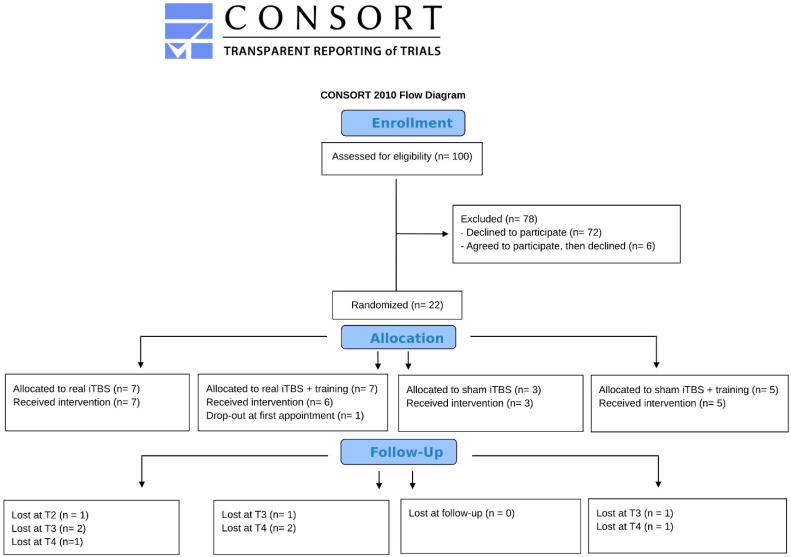
The CONSORT flowchart [78] includes the number of participants in each group and phase.

### 2.2. Outcome Measures

#### 2.2.1. Cognitive Impairment: The MATRICS Consensus Cognitive Battery (MCCB)

Cognitive functions were assessed using the MCCB, a consensus-derived cognitive assessment battery specifically built for schizophrenia [79,80]. The battery comprises 10 different tests assessing seven cognitive domains, namely the speed of processing (Trail Making Test A, Brief Assessment of Cognition in Schizophrenia—Symbol Coding, Category Fluency), attention/vigilance (Continuous Performance Test: Identical Pairs), working memory (Wechsler Memory Scale Spatial Span, Letter Number Span), verbal learning (Hopkins Verbal Learning Test), visual learning (Brief Visuospatial Memory Test), and reasoning/problem-solving (Neuropsychological Assessment Battery: Mazes). The test also includes an emotional intelligence subscale (Mayer–Salovey–Caruso Emotional Intelligence Test—Managing Emotions), which was excluded when computing the neurocognitive composite score.

#### 2.2.2. Standardized Clinical Scales

-The Italian version of the Brief Negative Symptom Scale (BNSS) [81,82] was used to evaluate negative symptoms. The scale measures five domains that are considered essential parts of the negative dimension according to the National Institute of Mental Health Consensus Development Conference [83]: affective flattening, alogia, anhedonia, avolition, and asociality. Higher scores on this questionnaire indicate greater negative symptoms.-The Positive and Negative Syndrome Scale (PANSS) [84] was administered to evaluate the severity of illness. It comprises 30 items that assess three major dimensions: positive symptoms, such as delusions, hallucinations, and suspiciousness (7 items); negative symptoms, such as blunted affect, emotional withdrawal, and lack of spontaneity (7 items); general psychopathology, such as anxiety, depression, and poor attention (16 items). The items are on a 7-point Likert scale, with higher scores indicating more severe symptoms.-The Calgary Depression Scale for Schizophrenia (CDSS) [85] was administered to assess depression symptoms. The scale includes nine clinician-rated items; higher scores indicate more severe depressive symptoms.-The Clinical Global Impression (CGI) [86] was completed by the clinician to assess current illness severity on a 7-point Likert scale, where higher rates indicate more severe illness.-The Specific Level of Functioning (SLOF) [87,88] was completed by caregivers or care workers to evaluate the patients’ behavioral functioning and daily living skills in self-care, social functioning, and community abilities. The higher scores indicate a better level of functioning.-The World Health Organization Quality of Life Assessment (WHOQOL-BREF) [89] assesses patients’ perceived quality of life. The self-administered questionnaire consists of 26 items measuring four domains related to the individual’s quality of life: physical health, psychological well-being, social relationships, and environment. Higher scores indicate a greater perceived quality of life.

Additionally, the assessment included the administration of tasks evaluating abilities of social cognition namely the Facial Emotion Identification Task [90], the Awareness of Social Inference Test [91], the Ambiguous Intentions Hostility Questionnaire [92], and the Mayer–Salovey–Caruso Emotional Intelligence Test—Managing Emotions (MSCEIT-ME) [93], which is part of the MCCB but can be excluded from the neurocognitive composite score. These tests have been previously described (and analyzed) elsewhere [74].

### 2.3. TMS Parameters

Stimulation was delivered through a Magstim Rapid2 magnetic biphasic stimulator connected to a 70 mm diameter figure-of-eight coil (Magstim Company, Whitland, UK).

In each stimulation session, iTBS was applied following the protocol described by Huang and colleagues [94].

This protocol is known for rapidly inducing a long-term potentiation process like synaptic plasticity [94,95]. It involves delivering 2 s trains of TBS (3 TMS pulses delivered at 50 Hz repeated every 200 ms) every 10 s (2 s stimulation and 8 s of intertrial interval).

Participants received 20 iTBS trains (600 total pulses—190 s) in each session. We adjusted the intensity of the stimulation to 100% of the active motor threshold (AMT) (mean intensity of 41.2 ± 5.7), which is defined as the lowest stimulator output intensity able to induce motor-evoked potentials with at least 100 µV of amplitude in the first dorsal interosseous muscle during an isometric contraction of 20% with a 50% probability [96]. 

Stimulation was delivered over the left DLPFC (10–20 EEG system: F3). We used the Softaxic Neuronavigation System version 3 (EMS, Bologna, Italy) and the Polaris Vicra infrared camera (NDI, Waterloo, Canada) to continuously monitor the coil position during the sessions. Pulses were delivered through a figure-of-eight coil held tangentially to the scalp with the handle pointing posteriorly. In the sham condition, we used the same coil placed 90° from the scalp. Participants received real or sham stimulation each working day for 3 weeks (15 total sessions). 

### 2.4. Cognitive Training

Computerized cognitive training was performed through the Cogpack software (version 9.3, Marker Software, Ladenburg, Germany). The program comprises exercises targeting domain-specific and non-domain-specific functions. Domain-specific tasks target individuals’ skills, such as verbal and visuospatial memory, working memory, executive functions, selective and sustained attention, and processing speed. The non-domain-specific exercises require a combination of several abilities, such as linguistic, mathematical, and basic logic skills. Each exercise typically includes different difficulty levels. Therefore, the training can be individually adjusted based on the patient’s baseline evaluation and improvements during the treatment sessions, aiming at enhancing the patient’s skills by increasing task difficulty and avoiding overly simple or excessively difficult exercises. 

Volunteers participated in a daily 50 min training session following the real/sham iTBS delivery. The first 30 min were dedicated to nonsocial cognitive training, whereas the other 20 min were spent on social cognition skills. Considering the nonsocial cognitive training, we created an exercise schedule covering the following functions over the three weeks: learning and memory, speed, working memory, attention, and executive functions (see Appendix A for the exercises’ details). The exercise schedule could be individually adjusted based on patients’ specific cognitive deficits and baseline abilities. For instance, some patients had some domains preserved, while others were more compromised. In this case, compromised functions were trained more (i.e., more repetitions) than preserved ones. The training was performed individually, with a trained psychologist who motivated the patients and discussed the strategies to solve the exercises to improve the patients’ metacognition.

The social cognition training took place after the exercises on nonsocial cognitive function. We used materials included in the emotion recognition and theory of mind modules of the Social Cognition Individualized Activities Lab (SoCIAL) [97,98] to train patients’ abilities to recognize emotions with static and dynamic stimuli and understand others’ mental states (see [74] for a detailed description). 

### 2.5. Procedure

After signing the informed consent, the baseline assessment took place. It included clinical assessments and was administered by a psychotherapist with neuropsychology expertise (S.T.). The evaluation was divided into two sessions to avoid patient fatigue. The clinical interview and social cognition battery were administered in the first session, whereas nonsocial cognitive abilities were evaluated in the second one. 

Participants were randomly assigned to one of the four groups and started the treatment the week after the baseline assessment. Participants were allocated using the RAND function in Excel based on the planned sample size of 40. Due to incomplete data collection, the group receiving sham iTBS with no training had a smaller sample size than the other groups. The study was conducted in a single-blind manner, meaning that participants were obviously informed about their training condition but unaware of whether they were receiving real or sham stimulation. None of the included participants had a previous experience with TMS. They underwent stimulation for 5 consecutive working days for 3 weeks. Each session was followed or not by the 50 min training depending on the assigned condition. Since iTBS has long-lasting effects, up to 60 minutes [99], we expected it to cover the time required for the entire training.

The assessment administered at baseline (T0) was repeated immediately after the three-week intervention (T1) and at three follow-ups, namely one month (T2), three months (T3), and six months (T4) after the intervention. We used parallel forms of questionnaires when available. After the T4 assessment, participants received a debriefing related to their assigned stimulation condition. They were then shown a graphical presentation of their performance at the different time points.

### 2.6. Primary and Secondary Endpoints

The primary aim of our study was to investigate the possibility of modulating negative symptoms and cognitive impairment through a combined intervention, time-locking iTBS with cognitive training. The primary endpoints, therefore, included pre–post scores on the BNSS and MCCB composite scores and subdomains. Secondary outcomes investigated possible delayed effects of the protocol at one, three, and six months after the end of the treatment. Baseline correlations between clinical, functional, and cognitive measures were explored. 

## 3. Statistical Approach

We analyzed the BNSS and MCCB scores in the R statistical programming environment [100] by applying linear mixed-effects models [101,102] using the LMER function of the lme4 package [103].

To analyze the primary endpoint, we added to the full model the fixed factors time (two levels: pre- vs. post-treatment), group (two levels: real vs. sham iTBS), and training (two levels: no training vs. training), and their interaction. The secondary endpoint analysis was different only for the fixed factor time that included five levels (pre- vs. post-treatment, plus the three follow-ups). We included the by-subject random intercept to consider the individuals’ variability. The inclusion of predictors in the final models was determined through a series of likelihood ratio tests (LRTs) in which we progressively removed the fixed factor that did not improve the overall model goodness of fit [104]. Details on the statistical approach and model selection are reported in the Appendix A, where additional references have been added ([105,106,107,108]). When interactions were significant, we performed post-hoc analyses using the phia package (testInteractions function) [109]. For the graphical presentation, the ggplot2 package was used [110]. For clarity, we report only the principal results in the main text, while the detailed analyses are presented in the Appendix A.

We ran correlations to explore the baseline relationships between the scales of functioning scores, clinical symptoms, and cognitive performance. Pearson correlation coefficients and two-tailed probabilities applying Bonferroni correction were computed. We plotted the correlation matrix using the corrplot package [111]. 

The dataset has been uploaded to a public repository (https://osf.io/ncvkt/ (accessed on 3 December 2023)). 

## 4. Results

### 4.1. Negative Symptoms

The analysis of the BNSS scores revealed a trend toward significance in the interaction between time and stimulation (χ^2^_(1)_ = 3.5, *p* = 0.060). The graphical representation (Figure 2—right panel) shows that such a trend can be explained by a reduction in scores after the iTBS treatment; however, the post-hoc analysis did not highlight significant differences between pre- and post-scores in the two stimulation conditions (*p*s > 0.168). Considering all data points, the null model was the best fitting one, not including fixed factors.

### 4.2. Cognitive Functions

Considering the MCCB composite score, the best fitting model included only the effect of time (χ^2^_(1)_ = 11.1, *p* < 0.001), with higher scores after the treatment compared to baseline. When adding the follow-up measurements, the effect of time was maintained (χ^2^_(4)_ = 36.4, *p* < 0.001), with a better performance in all post-treatment measurements compared to baseline (*p*s < 0.021) (Figure 3).

Considering the cognitive domains measured on the MCCB, we found an interaction between training and time (χ^2^_(1)_ = 3.8, *p* = 0.051) in verbal learning. Indeed, participants receiving the training improved after treatment compared to baseline (*p* = 0.016), while those not receiving it did not (*p* = 0.873). Crucially, no differences were found between the two groups at baseline (*p* = 0.600). When including follow-up measures, the interaction between training and time showed a trend (χ^2^_(4)_ = 7.9, *p* = 0.096). Planned comparisons suggested that verbal learning improved at the six-month follow-up compared to baseline only in the trained group (*p* = 0.006), while no differences were traceable in the group not receiving it (*p*s = 1) (Figure 4). 

Considering vigilance, the best fitting model included the interaction between training and stimulation (χ^2^_(1)_ = 3.7, *p* = 0.053). Post-hoc comparisons highlighted a trend between participants assigned to the training vs. no-training conditions in the sham group, with lower scores for the latter (*p* = 0.080). Considering all the data points, the interaction between training and stimulation remained significant (χ^2^_(4)_ = 5.9, *p* = 0.015), with participants assigned to the sham no-training condition having worse performance than the ones randomized to the sham training condition (*p* = 0.033) and the real stimulation no-training condition (*p* = 0.029). Moreover, the best fitting model included the interaction between training and time (χ^2^_(4)_ = 8.6, *p* = 0.072): participants assigned to the no-training conditions did not show improvement at the different time points (all *p*s > 0.077), while the ones assigned to the training showed better performance at T4 compared to T0 and T1 (*p* = 0.004 and *p* = 0.023, respectively) (see Figure 5).

### 4.3. Correlation Analyses

The correlation results reported here focused on the relationship between negative symptoms and cognitive scores with the functional and clinical scales (see [74] for detailed results between the clinical and functional scales). Negative symptoms measured through the BNSS positively correlated with scores on the three PANSS dimensions (positive, negative, and general psychopathology) and with the CGI, but negatively with the functional scale measured by caregivers or care workers (SLOF) and emotional intelligence measured through the MSCEIT (which was added as a representative measure of social cognition abilities [74]). A similar pattern emerged when considering the negative symptoms measured through the PANSS. The PANSS negative scale also negatively correlated with cognitive functions such as processing speed and working memory scores. Moreover, processing speed and visual learning were negatively correlated with the PANSS general psychopathology scale, whereas processing speed was positively correlated with the level of functioning. Problem-solving performance positively correlated with the years of education.

Considering the correlations among the cognitive functions, all but problem-solving and MSCEIT positively correlated with the MCCB composite score, and visual learning correlated with processing speed and verbal learning. Finally, the MSCEIT, which measures emotional intelligence, showed the already mentioned negative correlations with the three PANSS dimensions, the BNSS, and the CGI, as well as positive correlations with the SLOF and processing speed scores. Figure 6 represents the correlation matrix.

## 5. Discussion

In this pilot randomized controlled trial, our goal was to explore the potential of applying iTBS to enhance the effect of a personalized cognitive intervention in improving cognitive functions and negative symptoms in patients diagnosed with schizophrenia spectrum disorders. Indeed, it is well established that brain stimulation effects are state-dependent, meaning that stimulation interacts with the state of the targeted network, affecting brain activity and behavioral outcomes and possibly reducing interindividual variability in stimulation responses [65,66,67,68,69]. Since iTBS has been typically reported to increase cortical excitability [99] (but see [112]), we expected that priming the training with the stimulation of the left DLPFC would enhance the brain network activity and, in turn, maximize the impact of the training in improving cognitive functions and negative symptoms in patients assigned to the combined intervention. 

To this aim, we randomly assigned patients to four groups, in which we manipulated the behavioral treatment (no-training vs. training conditions) and the stimulation condition (real vs. sham iTBS). Clinical and cognitive outcomes were evaluated immediately after the treatment and at three follow-ups, one, three, and six months after treatment end. 

As stand-alone interventions, our findings suggest that iTBS and CR could improve negative symptoms and isolated cognitive functions such as verbal learning and vigilance, respectively, whereas the combined intervention does not support additional benefits to patients’ abilities. However, the limited sample of the current study prevents us from driving unambiguous conclusions, and the presented findings have a main descriptive value.

In line with this point, the first result that deserves discussion is the low acceptance rate of eligible patients to participate in the study. Although the COVID-19 pandemic waves certainly played a role in reducing the number of patients available to come to the hospital every day for three weeks, it is also true that we invited one hundred patients, and only twenty-two accepted and were subsequently randomized into the four groups. Patients declined the invitation primarily due to their lack of awareness concerning the illness and the feeling that they did not need any additional treatment. Secondly, the study was perceived as too demanding in terms of time and effort. Interestingly, we had only one drop during the intervention (the participant refused to continue after his first session), thus highlighting that the main challenge was engaging participants to start the treatment rather than continue it. Involving schizophrenic patients in clinical trials is a well-known issue, as they are more likely to refuse participation compared to patients with other psychiatric conditions [113,114]. The reason for this difference may be avolition, a key symptom of schizophrenia spectrum disorders, which plays a central role among negative symptoms and has been associated with poorer functional outcomes [8,115]. Future research, therefore, might consider running multicentric studies, using easily portable stimulation techniques such as transcranial direct current stimulation, or programming different session schedules to increase the possibilities of engaging schizophrenic patients in research treatments.

Considering the treatment effect on negative symptoms, the BNSS was preferred over the PANSS negative scale, which has been criticized due to the limited content validity and scoring reliance mainly on behavioral or performance deficits rather than internal experiences [116]. Conversely, the BNSS measures the five core negative symptom domains delineated in the 2005 NIMH Consensus Development Conference: anhedonia, avolition, asociality, alogia, and blunted affect [83].

Our findings showed a mild decrease in negative symptoms after iTBS, which was not confirmed at the follow-up evaluations. These results only partially aligned with previous studies. For example, a recent meta-analysis [117] suggested that iTBS over the left DLPFC reduced negative symptoms compared to the sham condition. Even meta-analyses on rTMS studies [51,52,53] supported a moderate effect of real stimulation in reducing negative symptoms. In contrast, a previous large-sample-size randomized multicenter clinical trial [118] showed no differences in negative symptoms between real and sham stimulation in a three-week protocol. Considering these inconsistencies, expert panel guidelines reduced recommendations in rTMS for treating negative symptoms from Level B, “probably effective” [119], to Level C, “possibly effective” [120]. The updated guidelines raised concerns about administering the PANSS in most of the revised studies and highlighted the patients’ heterogeneity across the trials, suggesting that it can contribute to the variability in the results. For instance, Aleman and colleagues [51] highlighted that individuals’ age and illness duration are potential moderators in predicting rTMS effectiveness, with stronger effects in reducing negative symptoms in younger patients with shorter illness duration. This represents an issue also considering the current findings: the heterogeneity in our sample may have prevented the emergence of more robust effects in the BNSS scores. Moreover, future research could consider combining cognitive interventions with other psychological treatments, such as cognitive behavioral therapy, which could improve the negative symptom outcomes [24,121,122].

Considering the effects of our intervention on cognitive abilities, patients assigned to the training condition showed a larger improvement only in verbal learning and vigilance abilities. The enhancement of verbal learning was evident after the treatment and at the six-month follow-up compared to baseline, thus highlighting long-lasting effects. Differences in the vigilance scores, instead, emerged only at T4 compared to the first two assessments. Findings on the composite MCCB score and the other subdomains (except visual learning) highlighted only a nonspecific effect of time, with higher scores after the treatment generally maintained for up to six months, probably reflecting exercise repetition improvements. Unlike our results, several previous meta-analyses highlighted the efficacy of CR in improving the trained functions even with small to moderate effect sizes [35,123,124]. More in line with our findings, verbal learning and attention/vigilance have been previously acknowledged as domains showing larger improvements than other cognitive functions [35,124,125,126]. Methodologically, several discrepancies between our study and previous ones must be considered, such as the number and frequency of CR sessions. For example, the meta-analysis by Cella and colleagues [123] investigated the effectiveness of CR in inpatients. The analyzed papers reported a mean of twenty-nine CR sessions (range 8–72). Vita et al. [35] tested several moderators, including treatment duration and the number of sessions per week. In this case, the average number of sessions was comparable to ours, but the distribution of the sessions differed, including an average number of 2.6 sessions per week. In our protocol, the short spacing between the training sessions, which took place every workday, could have influenced and weakened consolidation processes [127]. Considering our patients’ features, at present, it is unclear which individuals can benefit more from CR [128,129,130], with studies suggesting that more impaired patients are better candidates for the treatment and others pointing out that better cognition and lower severity at baseline may be associated with better outcomes [35,131]. 

Concerning the impact of iTBS, no effects were found in modulating participants’ performance in the cognitive tasks, nor as a stand-alone treatment or combined with the training. Previous studies showed inconsistencies considering the effect of stimulation on cognitive impairment, possibly due to the variability in stimulation protocols, the heterogeneity of patients, and the specific outcome measures [54,55,56,57]. Indeed, previous studies have focused on specific populations, such as early-phase psychosis [132], veterans [133], and treatment-resistant patients [134]. Finally, in some cases, the effects of rTMS on symptoms have been observed only weeks after the treatment [58,59,60], which have been previously interpreted as long-term cortical plasticity effects [135,136] and highlight the need for considering longer follow-ups in intervention studies including brain stimulation. 

Considering the relationship between clinical scales, negative symptoms, and cognitive performance, negative symptom severity was negatively correlated with processing speed and working memory, suggesting that higher negative symptoms largely impaired individuals’ performance in these functions. The association, however, was evident when negative symptoms were measured through the PANSS, but it was not replicated using the BNSS scores. This discrepancy is probably due to the criticism concerning the idea that the PANSS negative scale primarily measures behavior and performance referents rather than internal experiences [116]. No other correlations were present between nonsocial cognitive functions and negative symptoms, which is different from previous studies suggesting a frequent co-occurrence between the two [22,23]. The nature of the relationship between negative symptoms and cognitive impairment, however, remains to be established [137,138,139]. Larger correlations were found between the clinical scales measuring negative symptoms and the scale measuring emotional intelligence, an effect that has been inconsistently reported in the literature [140,141,142]. 

## 6. Limitations of the Present Study

The current study presents several limitations. The main one is the small number of participants included in the study. The COVID-19 pandemic and the many patients refusing to participate stopped us from collecting the expected sample size within the funding time constraints. The small sample size may also have consequences in terms of heterogeneity, which is high in this population and includes the presence of psychiatric and medical comorbidities, differences in illness duration and symptom severity that may play a role in the stimulation and training outcomes (for a recent review, see [35]). In our study, the four groups did not differ in demographic or clinical variables at baseline, whereas we did not assess the presence of psychiatric comorbidities or medical conditions (except those representing contraindications to stimulation). We did not select patients based on demographic or clinical features, but it remains an open question whether applying more restricted criteria would help clarify which participants would benefit more from the intervention. For example, including only first-episode schizophrenic or younger patients may increase the possibility of observing neuroplastic changes and, in turn, behavioral outcome modulations [143,144,145], but see [35] for different results. 

Last, we did not systematically check the effectiveness of stimulation condition blinding, a point that could be relevant considering the high placebo effect induced by non-invasive brain stimulation techniques (see, for example, [146,147]). 

## 7. Conclusions

To conclude, our data show that both iTBS and CR can effectively reduce negative symptoms and enhance isolated cognitive functions, whereas the combined intervention failed to boost patients’ improvements. Participants tolerated the stimulation well, and no major side effects were observed. This study includes several methodological strengths, considering using consensus-derived batteries to measure cognitive functions and negative symptoms. This choice is crucial to improving the assessment of clinical trial outcomes and reducing the heterogeneity of administering different tests across studies. The present study’s limitations, especially the small sample size, prevent us from making inferences on the current findings and providing straightforward conclusions but suggest, instead, the need for protocols able to engage patients to participate in research treatment. 

## Figures and Tables

**Figure 2 brainsci-14-00683-f002:**
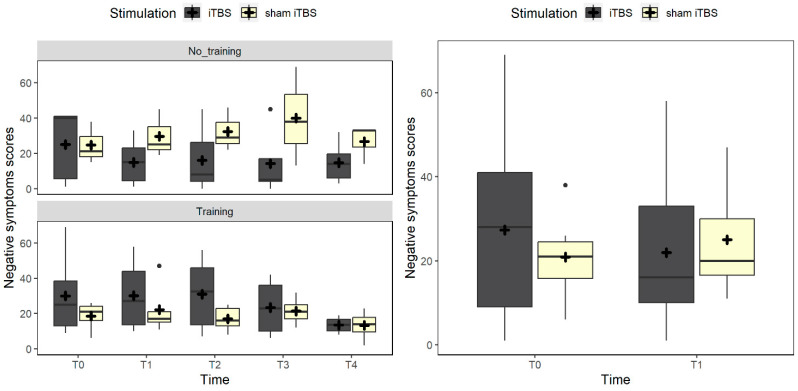
The figure represents the trend of negative symptom scores. On the left panel, the box plots compare real vs. sham iTBS in the no-training and training conditions. The right panel shows the pre–post change in negative symptoms in the real vs. sham iTBS conditions. Black dots represent outliers, and the cross symbol represents the mean values.

**Figure 3 brainsci-14-00683-f003:**
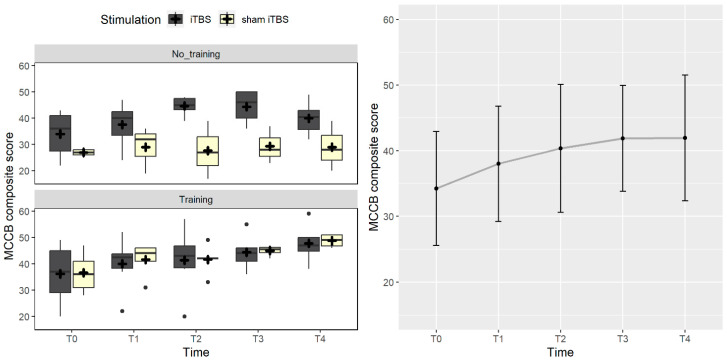
The figure represents the MCCB composite score trend. On the left panel, the box plots compare real vs. sham iTBS in the no-training and training conditions. The right panel shows the main effect of time. Black dots represent outliers, and the cross symbol represents the mean values.

**Figure 4 brainsci-14-00683-f004:**
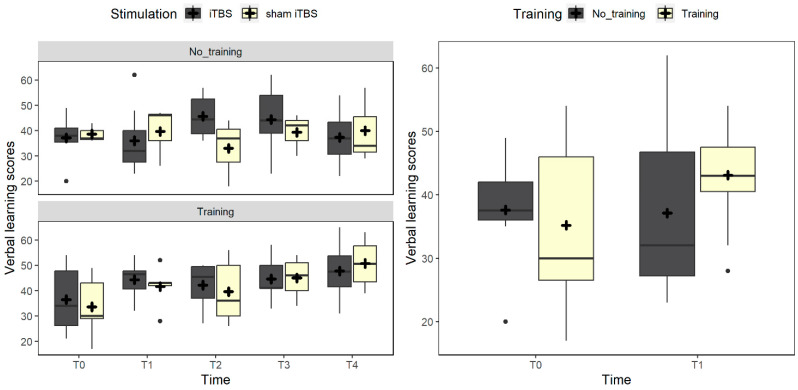
The figure depicts the verbal learning scores trend. On the left panel, the box plots compare real vs. sham iTBS in the no-training and training conditions. The right panel represents pre–post changes in the conditions of no training (gray box) vs. training (light yellow). Black dots represent outliers, and the cross symbol represents the mean values.

**Figure 5 brainsci-14-00683-f005:**
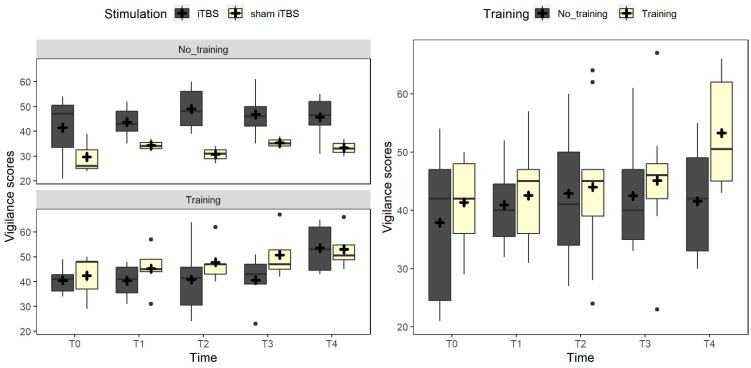
The figure represents the vigilance scores trend. On the left panel, the box plots compare real vs. sham iTBS in the no-training and training conditions. The right panel represents the trend at the different time points in the conditions of no training (gray box) vs. training (light yellow). Black dots represent outliers, and the cross symbol represents the mean values.

**Figure 6 brainsci-14-00683-f006:**
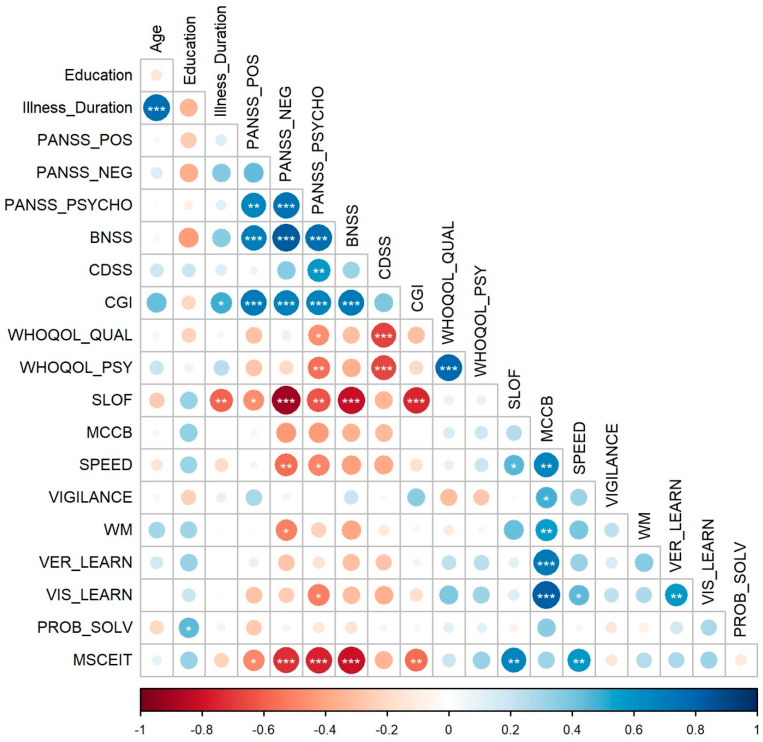
The figure represents the correlation matrix displaying the baseline relationships between each variable pair. The red and blue colors indicate negative and positive correlations, respectively. The intensity and size of the dots are proportional to the correlation coefficients, and asterisks inside the dots denote statistical significance. (* *p* < 0.05, ** *p* < 0.01, *** *p* < 0.001). Note: BNSS = Brief Negative Symptoms Scale; CDSS = Calgary Depression Scale for Schizophrenia; CGI = Clinical Global Impression; MCCB = MATRICS Consensus Cognitive Battery composite score; MSCEIT = Mayer–Salovey–Caruso Emotional Intelligence Test; PANSS_NEG = Positive and Negative Syndrome Scale (PANSS) negative symptoms; PANSS_POS = PANSS positive symptoms; PANSS_PSYCHO = PANSS general psychopathology; PROBL_SOLV = problem-solving score; SLOF = Specific Level of Functioning; SPEED = processing speed score; VER_LEAR = verbal learning score; VIGILANCE = attention/vigilance score; VIS_LEAR = visual learning score; WHOQOL_QUAL = World Health Organization Quality of Life Assessment—quality score; WHOQOL_PSY = World Health Organization Quality of Life Assessment—psychological well-being; WM = working memory score.

**Table 1 brainsci-14-00683-t001:** Baseline demographic and clinical characteristics of participants (means ± SD).

Variable	Sham (*N* = 3)	iTBS (*N* = 7)	Sham + Training (*N* = 5)	iTBS + Training (*N* = 6)
**Demographic**				
Gender (f/m)	2/1	2/5	0/5	1/5
Age	32.3 ± 8.3	38.3 ± 11.2	31.2 ± 8.9	36.3 ± 13.8
Education	12.3 ± 1.2	12.3 ± 1.9	11.4 ± 2.1	12.2 ± 2.0
Illness duration (years)	4.7 ± 0.6	11.1 ± 10.7	7.6 ± 4.3	13.3 ± 11.0
**Diagnosis**				
Schizophrenia	2	5	3	4
Schizoaffective disorder	1	1	1	1
Psychotic disorder NOS	-	1	1	1
**Clinical measures**				
PANSS	70.3 ± 6.5	62.6 ± 22.7	53.6 ± 8.0	71.0 ± 24.8
BNSS	24.7 ± 11.9	25 ± 19.7	18.6 ± 8.0	30 ± 22.7
CDSS	8.0 ± 5.3	5.3 ± 5.0	4.4 ± 1.1	6.2 ± 3.4
CGI	3.7 ± 0.6	3.6 ± 1.5	2.8 ± 0.8	3.8 ± 1.5
SLOF	165.3 ± 27.2	185 ± 24.5	193.2 ± 15.7	166.5 ± 36.1
WHOQOL ‘quality’—item G1	3.7 ± 0.6	3.16 ± 1.5	3.8 ± 0.4	3.8 ± 1.0

Notes: BNSS = Brief Negative Symptom Scale, CDSS = Calgary Depression Scale for Schizophrenia, iTBS = intermittent theta burst stimulation, CGI = Clinical Global Impression, NOS = not otherwise specified, PANSS = Positive and Negative Syndrome Scale, SLOF = Specific Level of Functioning, and WHOQOL = World Health Organization Quality of Life Assessment.

## Data Availability

The dataset has been uploaded to a public repository (https://osf.io/ncvkt/ (accessed on 3 December 2023)).

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
