# Peer review of "Intermittent Theta Burst Stimulation Combined with Cognitive Training to Improve Negative Symptoms and Cognitive Impairment in Schizophrenia: A Pilot Study"

_brainsci, 2024, doi:10.3390/brainsci14070683_

Round 1

Reviewer 1 Report

Comments and Suggestions for Authors

Brief summary: This interesting article focuses on the study of patients suffering from schizophrenia. The authors combined intermittent theta burst stimulation with cognitive remediation to evaluate potential improvements in negative symptoms and cognitive impairement. They conducted the experiment on 21 patients that were randomized into four groups. They reported marginal modulation of the intermittent theta bursts on negative symptoms and no effect on cognitive functions.

Please see my comments below.

Introduction:

- Positive symptoms are introduced very vaguely as well as negative symptoms (lines 40-41). Please use current relevant literature to better define these concepts (i.e: positive symptoms are mostly hallucinations and delusions, whereas negative symptoms are well defined in the DSM-5 over several components such as Alexythimia, Amotivation etc.)

- It would be pertinent for the readership to specify the cognitive deficits mostly found in individuals with schizophrenia rather than providing the broad definition of cognitive deficits. There is a vast array of literature on this topic that could be relevant to bonify the elements presented over lines 42-47.

- Major cognitive treatment approaches for schizophrenia are not discussed such as cognitive behavioral therapy. This could be discussed further over lines 52-54. 

- Along these lines, it could be beneficial for the readership to have concrete exemples of cognitive remediation techniques considering this concept is central to the study.

- It is confusing that the introduction of the DLPFC (line 67) comes after the explanations on studies about the use of rTMS considering that rTMS targets DLPFC in most cases. Discussing brain areas related to cognitive impairment and negative symptoms could be introduced sooner in the introduction to account for clarity.

- The aim of the study could be more explicit to the readership and the hypothesis should be discussed. The authors stated the methodology used (line 82), but the objective remain unclear. 

Materials and methods:

- How was power analysis conducted? (i.e why 100 patients were initially invited?)

- How were the participants recruited?

- Considering that mental health comorbidities were not included as exclusion criteria, how was comorbidity (which is usually the norm in this population) handled to interpret the results?

- Please provide the psychometric information on the selected scale to justify their use (i.e: external validity and reliability metrics). A table form could account for clarity.

- From the methodology, statistical significance and data analysis remain unclear. How was the statistical model selected and why was this model pertinent to the data that was collected?

Results:

- Results are, in my opinion, well presented and visiually appreciable.

Discussion:

- Study limitations are overlooked. The authors report solely the small number of participants. Please include a sub-section of the discussion on the limitations of the study and include relevant limitations related to the model used.

- It would be interesting for the readership if the conclusion had its own section to better highlith the results of this study. 

- A small comment on line 405: nothing in the presented study evaluated the feasibility of this approach, therefore claiming that this study show the feasibility of study (especially considering that this was not the aim and nothing evaluated the feasbility in the methodology) is too broad. Please clarify.

Minor comments:

- Please follow MDPI's guidelines on referencing style. As an exemple  [1], [2] should be  [1,2].

- rTMS and TMS are not the same thing (lines 61-62). rTMS received considerable attention, not simply TMS. Please clarify.

Comments on the Quality of English Language

Nil

Reviewer 2 Report

Comments and Suggestions for Authors

The manuscript "Intermittent theta burst stimulation combined with cognitive training to improve negative symptoms and cognitive impairment in schizophrenia: a pilot study" is a well-written pilot study investigating 3-weeks rTMS on a small number of patients with a diagnosis of schizophrenia.

The authors provide their data (excellent) and the supplementary material provides the statistical analyses (very nice, thank you).

I have one major concern, and a few minor points.

The major concern regards the choice of GLM given the small N. You would need at least n=10 per group.

Full article: Power and Sample Size for Fixed-Effects Inference in Reversible Linear Mixed Models (tandfonline.com)

I also could not find a test of assumptions. With such small N you likely need to use non-parametric tests (your boxplots nicely show that data is not normally distributed). 
Friedman's ANOVA is more appropriate with small N

minor issues: 

line 62: rTMS not established (repetitive TMS)

line 73: CR not established

line 84: iTBS not established

line 97: I think DS = 8.8. should be SD = 8.8

table 1: was illness duration significantly different between the 4 groups? The two sham groups have smaller mean and SD. If so, did you consider using it as a covariate?

section TMS: did you perform any debriefing? If so, when?
if not, did you at least ask them about discomfort / side-effects and or what they guessed in which group they were

line 181: states 30 min whereas on line 207 it states 50 min. Please clarify

Figures 2-5: since you use boxplots the line is the median and the cross is - I guess - the mean. The legends should be amended by stating that the cross represents the mean

result section: please report effect sizes, Cramer's V for chi-square test. For small N p-values are often not significant, but the effect size can still be in the medium range

line 303: showed and already <- should be "showed an already"

fig 6: purely personal taste, but from the weather forecast I am used to red = positive / warm and blue = negative / cold values. You have the opposite. No big deal, you can keep it if you prefer

discussion section: it is more common to first summarise the results / main findings before discussing it, i.e. move line 350ff up and before the sentence starting on line 324

line 389 plastic should be plasticity   

Round 2

Reviewer 1 Report

Comments and Suggestions for Authors

This is the second iteration of revision for the manuscript entitled '' Intermittent theta burst stimulation combined with cognitive training to improve negative symptoms and cognitive impairment in schizophrenia: a pilot study ''.

Considering the quality of the revision made by the authors and the detailed responses to my previous comments, I have no further comments.

Comments on the Quality of English Language

Nil